# Does Adjustment of Antiseizure Medication Regimen after Failed Epilepsy Surgery Improve Outcomes?

**DOI:** 10.3390/medicina59040785

**Published:** 2023-04-18

**Authors:** Yuxin Wu, Zaiyu Zhang, Ping Liang, Lusheng Li, Bin Zou, Difei Wang, Xinyu Dong, Haotian Tang, Hanli Qiu, Xuan Zhai

**Affiliations:** 1National Clinical Research Center for Child Health and Disorders, Ministry of Education Key Laboratory of Child Development and Disorders, Department of Neurosurgery, Children’s Hospital of Chongqing Medical University, Chongqing 400015, China; 2021150180@stu.cqmu.edu.cn (Y.W.); mr_zaiyuzhang@163.com (Z.Z.); liangping868@sina.com (P.L.); lilusheng@cqmu.edu.com (L.L.); zoubin2017@163.com (B.Z.); wangdifei1993@163.com (D.W.); seizures@163.com (X.D.); tang_haotian_0996@163.com (H.T.); hanliqiu_1207@163.com (H.Q.); 2Chongqing Key Laboratory of Translational Medical Research in Cognitive Development and Learning and Memory Disorders, Chongqing 400015, China

**Keywords:** epilepsy, epilepsy surgery, antiepileptic drug, surgery failure, quality of life

## Abstract

*Background and Objectives:* After failed epilepsy surgery, patients often revert to an antiseizure medication (ASM) ASM regimen, which can be adjusted or optimized in three ways: increasing the dose, alternative therapy, and combination therapy. It is unclear which type of antiseizure medication adjustment method can improve outcomes. *Materials and Methods*: Children who underwent failed epileptic resection surgery at the Department of Neurosurgery, Children’s Hospital of Chongqing Medical University between January 2015 and December 2021 were included in this cohort, who were reviewed for whether they underwent adjustment of ASM with increased dose, alternative therapy, or combination therapy. The seizure outcome and quality of life (QoL) were assessed. Two-tailed Fisher exact test and Mann–Whitney U test were used for statistical analysis. *Results*: Sixty-three children with failed surgery were included for further analysis, with a median follow-up time of 53 months. The median seizure recurrence time was 4 months. At the last follow-up, 36.5% (*n* = 23) of patients achieved seizure freedom, 41.3% (*n* = 26) achieved seizure remission, and 61.9% (*n* = 39) had a good QoL. None of the three types of ASM adjustment improved children’s outcomes, whether considered in terms of seizure-free rate, seizure remission rate, or QoL. Early recurrences were significantly associated with decreased probability of seizure freedom (*p* = 0.02), seizure remission (*p* = 0.02), and a good QoL (*p* = 0.01). *Conclusions*: Children who underwent failed epilepsy surgery remains some potential for late seizure remission from ASM. Yet adjusting ASM regimen does not increase the probability of seizure remission nor does it improve the QoL. Clinicians should complete evaluations and consider the need for other antiepileptic treatment as soon as possible after surgery failed, especially when dealing with children with an early recurrence.

## 1. Introduction

Epilepsy is one of the most common neurological diseases affecting more than 70 million people in the world [1,2]. About 1/3 of patients cannot control seizures through antiseizure medication (ASM) and suffer from drug-resistant epilepsy. Surgery is the evidence-based treatment for children with drug-resistant partial seizures, with 50–80% of children achieving seizure freedom postoperative [3]. However, approximately 30% of children still have residual seizures after surgery [4,5]. Residual seizures are the most important factor affecting the quality of life (QoL) in children with epilepsy [6], and it is known that persistent seizures increase disability, mortality, cognitive performance declines, and psychosocial problems [7]. Therefore, it is important to control residual seizures in children who have failed surgery.

For clinicians, the management of patients after the failure of initial surgery is challenging [8,9]. Depending on the seizure types, seizure frequency, and pathological findings, individualized therapy is required. Clinicians are faced with a variety of treatment options: medication adjustment, secondary surgery, vagus nerve stimulation (VNS), ketogenic diet, etc. [7]. ASM regimen adjustment may be the first choice for most clinicians, which can be categorized as increasing the dose, alternative therapy, and combination therapy [10]. The development of a postoperative ASM regimen relies on clinician experience, and there is a lack of evidence for effective ASM adjustment methods. Previous studies have focused on the effectiveness of secondary surgery, VNS, or other new treatments after surgical failure, and few studies have investigated the role of the ASM regimen in children with residual seizures [8].

Therefore, the aim of this study was to explore which method of ASM regimen adjustment was better at improving the seizure outcome or QoL and to discuss subsequent treatment for children after the failure of initial surgery.

## 2. Materials and Methods

### 2.1. Study Population

From January 2015 to December 2021, 283 children (<18 years old) underwent epilepsy resection surgery at the Department of Neurosurgery, Children’s Hospital of Chongqing Medical University, Chongqing, China. In the cohort, 65 children who had failed surgery were included for further analysis. Some of these data were reported in our previous article on QoL of children with residual seizures after surgery [11]. The inclusion criteria were as follows: 1. children who underwent failed epileptic resection surgery; 2. at least 1 year of follow-up after seizure recurrence; 3. all patients suffered from drug-resistant epilepsy defined as persistent seizures after at least 2 unsuccessful antiepileptic medication regimens. The definition of failure of epilepsy surgery varied across studies. Some studies used stricter criteria, regarding failing to achieve seizure-free (Engel classification 1A) as surgical failure, and some studies considered moderate or no improvement in seizures as failure [12]. The definition of surgical failure adopted in this study was the recurrence of seizures after surgery (Engel classification II–IV), except for acute seizures within 2 weeks after surgery and seizures associated with the withdrawal of ASM.

The exclusion criteria were as follows: 1. children underwent hemispherectomy and palliative surgery (e.g., neurostimulation surgery, corpus callosotomy); 2. epilepsy syndromes or epileptic encephalopathy; 3. died during the follow-up period; 4. if children received other antiepileptic therapy such as secondary surgery, VNS, or ketogenic diet during the follow-up period, the follow-up endpoint was defined as the time the child received the other antiepileptic treatment, and the outcome was assessed at that time point.

This study was approved by the Ethics Committee of Children’s Hospital of Chongqing Medical University. All patients or their families were informed about the study and were given signed consent forms allowing the use of patients’ medical record data for the study.

### 2.2. Postoperative Management

Children return to the clinic every 3 months after surgery. After a seizure recurrence, there are four options for ASM therapy, including continuing the current ASM therapy, increasing the dose, alternative therapy, and combination therapy. Increasing the dose means that the child is still taking the current ASM, but the dose is increased based on body weight. Combination therapy refers to the addition of other types of antiseizure medication. Alternative therapy refers to switching from one ASM to another without changing the total number of ASMs. The ASM regimen is based on an individual medical basis (ASM already tried, seizure type, seizure frequency, age, adverse effects, ASM mechanism, patient preference, and other factors) at the discretion of the neurologist.

### 2.3. Data Collection

Data were obtained from electronic medical records. Seizure frequency was classified as daily, weekly, monthly, or less. Surgery site was divided into temporal and extratemporal lobe regions. Early recurrence was defined as seizure recurrence within 6 months after surgery. Seizure outcome was assessed using the Engel seizure classification. The efficacy of ASM regimen adjustment methods was assessed according to three criteria, including seizure freedom (Engel classification I); seizure remissions (Engel classification I-II); and quality of life. QoL was assessed using the health-related quality of life measure for children with epilepsy (CHEQoL-25). As suggested by the authors of this questionnaire, there are 2 categories of QoL: good (>60) and poor (<60) [13].

### 2.4. Data Analysis

Data were analyzed using SPSS 27.0 software. Continuous variables were summarized as means ± standard deviations, and categorical variables were expressed as counts (percentages). The association between outcomes and categorical variables such as gender, surgical site, and ASM regimen adjustment method was analyzed using chi-squared test or Fisher exact test. The association between outcomes and duration of epilepsy was analyzed using the Mann–Whitney U test. As an individual with residual seizure after surgery may successively undergo dose increasing, alternative therapy, and combination therapy, it is not possible to simply divide children into 4 groups: continuing the current ASM therapy, increasing the dose, alternative therapy, or combination therapy. For this reason, we chose a between-group comparison approach for our statistical analysis, comparing the outcomes of children with and without ASM adjustments, comparing the outcomes of children with and without dose increases, and so on.

## 3. Results

Sixty-five children were included in the cohort, of which two patients were excluded from the analysis cohort due to death during follow-up. The median follow-up after surgery was 53 months and the median follow-up after seizure recurrence was 47 months. The 63 children were approximately equally divided between males and females, with 57.1% males and 42.9% females. The mean duration of epilepsy was 17.1 ± 25.7 months. All 63 children experienced surgical failure, with a mean time to seizure recurrence of 4.8 ± 3.4 months after surgery, and 66.7% (*n* = 42) had an early recurrence. A total of 41.3% (*n* = 26) were admitted for brain tumors, 17.5% (*n* = 11) for focal cortical dysplasia II, 12.7% (*n* = 8) for focal cortical dysplasia I, 11.1% (*n* = 7) for vascular malformations and 7.9% (*n* = 5) for encephalomalacia focus. Detailed data are shown in Table 1. Due to sample size limitations, we did not analyze the effect of different pathological types on the outcome of children with surgical failure. See Appendix A for additional data.

### 3.1. Postoperative Management

Of the 63 patients who failed surgery, 23 (36.5%) continued their previous ASM regimen after seizure recurrence, while 40 (63.5%) adjusted their ASM regimen under medical supervision. The most common method of ASM adjustment was dose increasing (*n* = 38), followed by combination therapy (*n* = 17), and only 11 patients tried alternative therapy. With antiseizure medication, 23 (36.5%) of children achieved seizure freedom at follow-up, 26 (41.3%) achieved seizure remission and 39 (61.9%) had a good QoL. Three (4.8%) patients underwent a second operation after unsatisfactory ASM therapy, and all were seizure-free after the second operation.

### 3.2. The Relationship between ASM Regimens and Outcomes

The seizure-free rate was 47.8% and 30.0% for children who adjusted the ASM regimen and children who did not adjust the ASM regimen, respectively, with no statistically significant difference in the distribution between the two groups (chi-square = 2.00, *p* = 0.16); the seizure remission rate was 52.2% and 35.0% for children adjusted ASM regimen and children did not adjust ASM regimen, respectively, with no statistically significant difference in the distribution (chi-square = 1.78, *p* = 0.18); there was also no statistical difference in QoL between the two groups (chi-square = 0.17, *p* = 0.68). Neither dose increase, alternative therapy, nor combination therapy improved seizure outcomes or QoL, as detailed in Table 2. In the between-group comparison, children with late recurrences were more likely to be seizure-free (*p* = 0.02), to achieve seizure remission (*p* = 0.02), and to have a good QoL (*p* = 0.01) than children with early recurrences. In addition, children with complete resection of the epileptic focus were more likely to achieve seizure remission (*p* = 0.04).

## 4. Discussion

With the rapid advances in epilepsy surgery techniques, epilepsy surgery is generally considered to be the last hope for drug-resistant patients who may not be able to bear the loss of that hope if surgery failed [14]. At this point, clinicians should complete an individualized assessment as soon as possible to determine what additional treatment is needed, including secondary surgery, VNS, etc. [8]. However, not all patients who have failed surgery would follow their doctor’s recommendation for additional antiepileptic treatment, for reasons of individual preference or other considerations. Instead, many patients will first request additional ASM to control their seizures [14]. Clinicians have four options regarding ASM regimens: continuing with the current regimen, increasing the dose, alternative therapy, and combination therapy. However, previous studies have not shown the significance of ASM regimen adjusting on seizure outcomes, nor have they clarified which adjustment method is better. The present study retrospectively included 63 children who had failed epilepsy surgery to investigate whether adjustment of ASM regimens could provide benefits in terms of seizure control or quality of life, and which adjustment method specifically improved patient outcomes the most. We found that adjustment of ASM did not improve the probability of seizure freedom, nor did it improve patients’ seizure outcomes or QoL. Furthermore, neither dose increases, alternative therapy nor combination therapy improved the outcomes of patients with surgical failure in terms of seizure freedom, remission rates, or QoL in patients with surgical failure.

After seizure recurrence, in a proportion of children, seizures will eventually resolve after months or years, referred to as late seizure remission. In other cases, the seizures will continue indefinitely [12]. Subsequent treatment after failed epilepsy surgery is challenging, partly because in most cases the reasons for the failure of epilepsy surgery are unknown [15]. In a small retrospective study, initial surgical failure was attributed in 35% of cases to functional considerations, in 35% to the incorrect delineation of the epileptic zone, and in 30% to failure to perform the resection as originally planned [16]. On the other hand, it is because of the diversity of treatment options and the lack of strong evidence as to which option is better. In previous work, clinicians usually adjusted ASM based on clinical experience, with a strong subjective element.

Residual seizures after epilepsy surgery should raise concern, bearing in mind that failure of epilepsy surgery does not mean that these children do not have the potential to achieve seizure freedom [17]. Although children did not respond well to ASM preoperative, and were all diagnosed with drug-resistant epilepsy, between 15% and 38% of patients who initially failed epilepsy surgery achieved late seizure remission with post-surgical support based on ASM [18]. One plausible explanation is that although epilepsy surgery did not immediately make patients achieve seizure-free, it may increase the responsiveness of the epilepsy network to ASM [18]. The postoperative management of these children who have failed epilepsy surgery is therefore important and challenging, but there is a paucity of data in the literature on the management, and few studies have examined the role of postoperative medical management [8,18,19]. In clinical practice, neurologists usually adjust ASM according to the patient’s seizure characteristics, patient preference, etc. The postoperative ASM regimen depends on the experience and subjective judgment of the neurologist, and there is a lack of evidence-based management options.

Ma J et al. analyzed a cohort of 103 adults who failed surgery and found that a small proportion (27.2%) of patients benefited from ASM adjustment; however, the vast majority of patients (72.8%) did not benefit from ASM adjustment [8]. Janszky J et al. reviewed seizure outcomes at 2 years postoperatively in 86 patients who underwent failed temporal lobe epilepsy surgery and found no positive impact of ASM regimen changes on postoperative seizure control [19]. In this study, which adds to the breadth of outcome assessment based on previous studies, we assessed the value of ASM modifications for seizure remission and quality of life. We found no significant differences in late seizure-free rates, seizure remission rates, or quality of life between children with and without ASM adjustments after surgery failure when specific antiepileptic drug classes were not considered. In addition, we reviewed the way post-operative pharmacological regimen adjustment in more detail, subdividing them into dose increasing, alternative therapy, and combination therapy. We explored whether there is any particular type of ASM regimen adjustment that may be helpful in terms of seizure control or quality of life. This is an important but under-discussed topic in the previous literature. Only a small number of single-center retrospective studies have discussed a specific type of ASM adjustment, but the systematic analysis is lacking. Salanova V et al. found that of 20 patients with unsatisfactory postoperative seizure outcomes, only two achieved seizure freedom after the introduction of a new ASM [20]. This study analyzed the different ASM modification methods in more detail and found that neither dose-increasing, alternative therapy nor combination therapy improved seizure outcomes or quality of life in children who had failed epilepsy surgery. Therefore, we recommend that post-operative children should try additional antiepileptic treatments such as a ketogenic diet and neurostimulation early in case of seizure recurrence. The ketogenic diet is considered almost a last resort in the treatment of refractory epilepsy after many types of ASMs and even epilepsy surgery has failed. In fact, it has long been suggested that all children with epilepsy could try a ketogenic diet before trying other options, once the relevant contraindications have been ruled out [21]. Neurostimulation surgery has also been proposed for patients who have failed initial surgery [22]. Vale FL et al. found that in patients with postoperative seizures, VNS reduced seizure frequency by 30% in 35% of patients and improved QoL in nearly half of patients [23]. A recent meta-analysis showed that the 50% responder rate (50% RR) and seizure freedom rate after VNS were 56.4% and 11.6%, respectively, and that fewer types of ASM tried before VNS were associated with better seizure outcomes after VNS implantation [24]. As all methods of ASM adjustment do not lead to better seizure outcomes, and early VNS (fewer ASM trials) may be a modifiable factor in the ideal seizure outcome, VNS before repeated trials of other ASMs may be a more appropriate option for children who have failed surgery. Given the small sample size of this study, the results should be interpreted with more caution. The seizure freedom rates in this cohort were 47.8% and 30.0% for children with and without ASM adjustment, respectively, and there appears to be some difference between the seizure freedom rates in the two groups, although not statistically significant (*p* = 0.16). Furthermore, Kaye LC et al. found that patients who underwent ASM adjustment appeared to have a significant benefit more than 5 years after surgery [18]. The median follow-up of our cohort was 53 months, and it is possible that children who experienced ASM adjustment achieved seizure freedom during longer follow-ups. Therefore, we cannot exclude the possibility that a different conclusion may be reached after expanding the study cohort and longer follow-up. Thus, a multicenter, prospective study is needed to further confirm our conclusions. It is important to note that this study does not deny the value of ASM management for patients with persistent postoperative seizures, rather we believe that ASM management may still play a key role. The focus of this study is that clinicians should combine other antiepileptic treatments as early as possible after surgical failure, rather than repeatedly adjusting ASM, which may not result in significant patient benefit.

Secondary surgery is also a topic of interest. A meta-analysis showed that after failed epilepsy surgery, the seizure-free rate after secondary surgery was 47% [25], and secondary surgery appears to be particularly beneficial for some children with refractory epilepsy secondary to cortical dysplasia [16]. This is an exciting result. For patients who have failed the initial surgery, a second surgery is certainly the best way to control seizures. However, secondary surgery was not a common choice at our center. Only 3/63 patients underwent secondary surgery, although all three patients had good seizure outcomes. Firstly, secondary surgery appears to be a valuable option but is not appropriate for all patients, such as those with evidence of multiple epileptogenic focus, poor localization, or multifocal EEG abnormalities are not suitable for secondary surgery. Secondly, the high risk of complications also needs to be weighed up when considering additional surgery [26]. An increased incidence of permanent neurological deficits, overall surgical complications, and visual field deficits due to secondary surgery have been reported [27]. Thirdly, secondary surgery in a group of patients who have failed the initial procedure is challenging for a variety of reasons. Postoperative changes such as scar formation, including adhesions of the dura to vital cortical veins and adhesions of medial temporal lobe structures to arteries, cranial nerves, and brainstem structures, may complicate the procedure. Scalp EEG may also be difficult to interpret due to cranial defects, dural scarring, displacement of cranial cavities, and anatomical structures [28]. For these reasons, secondary surgery is not commonly performed at our center. However, when indicated, secondary surgery should be performed as early as possible to benefit from greater functional plasticity and to compensate for neurological deficits. The preoperative preparation needs to be more careful and thorough than the initial surgery.

The study also found that children with late recurrences were more likely to be seizure-free, get seizure remission, and have a good QoL than those with early recurrences (at a 6-month cut-off). In previous studies, the definition of early recurrence varied, but Goellner E et al. found that 6 months was the optimal threshold and that patients with early recurrence had worse outcomes, required higher numbers of ASM, and underwent secondary surgery more frequently [27,28]. Najm I et al. also found that patients with early recurrence were less likely to respond to ASMs. In contrast, late recurrence tended to be milder, less frequent, and more easily controlled by ASM adjustments. In addition, late epilepsy is most seen in patients without clear pathological findings [29]. Ramesha K. N. et al. find that seizure recurrence within 1 year after surgery increases the risk of subsequent seizures by four to seven times [30]. Patients with late recurrences usually have fewer seizures and a better QoL compared to individuals with early recurrences [31]. A plausible explanation is that the epileptogenic tissue is not “static” and can be affected after surgical perturbation of the brain network that includes the epileptogenic region. Early recurrences are usually due to errors in localization and/or resection of epileptogenic foci, whereas late recurrences may be due to the development/maturation of new and active epileptic foci [29], Some MRI post-processing techniques have detected abnormal lesions that were missed in the initial assessment, laterally confirming this speculation [12]. Therefore, in the long term, the different characteristics of early and late recurrence may have implications for the planning of treatment options [28]. Since patients with late recurrences are more likely to benefit from ASM, more trials should be completed to optimize ASM regimens for seizure control in children with late recurrences. Additionally, in children with early recurrence, it may be necessary to add other anti-epileptic treatments at an earlier stage.

The present study has several limitations. Firstly, we did not record the time between seizure recurrence and return to seizure-free status because this is a cross-sectional study, and a retrospective investigation of this data would likely introduce a large recall bias. Yet these data are crucial because they can help determine when it is most appropriate to stop adjusting ASM and start other antiepileptic treatments. Secondly, this study did not consider the effect of the type of antiseizure medication, which was limited by the sample size of our center. Thirdly, we did not assess the tolerability of the ASM protocol and patient compliance with the ASM. Fourthly, we did not analyze the effect of different pathological findings on outcomes. It has been reported that patients with cavernous hemangioma are more likely to achieve late remission of seizures, so we cannot rule out the possibility that completely different conclusions may be reached in patients with certain pathological findings [18]. Finally, the small sample size of this study reduces the credibility of the findings. Despite these limitations, we believe that this study is innovative because it considers outcomes in terms of seizure freedom, seizure remission, and QoL. Additionally, all children were followed for more than 2 years, and to our knowledge, it is the first to provide evidence that increased dose, alternative therapy, and combination therapy do not improve outcomes in patients who have failed surgery.

## 5. Conclusions

There is still potential for late seizure remission with ASM after failed epilepsy surgery in children. There were no statistical differences in seizure-free rates, seizure remission rates, or quality of life between children who adjusted their antiepileptic medication and those who continued their current antiepileptic medication. Dose increasing, alternative therapy and combination therapy do not improve seizure outcomes or quality of life. Children with late recurrences have better seizure outcomes and higher quality of life than those with early recurrences. Clinicians should perform an assessment soon after seizure recurrence and consider whether additional antiepileptic treatment is needed, especially in patients with early recurrences.

## Figures and Tables

**Table 1 medicina-59-00785-t001:** Characteristics of children with failed resection surgery.

Characteristic (*n* = 63)	Number of Patients (%)/ Mean ± SD (Range)
Gender	
Male	36 (57.1%)
Female	27 (42.9%)
Duration of epilepsy, months	17.1 ± 25.7 (1–110)
Seizure type	77.9 ± 37.6 (18–192)
Focal seizure	36 (57.1%)
Generalized seizure	27 (42.9%)
Surgical site	58 (48.7%)
Temporal	22 (34.9%)
Extratemporal	41 (65.1%)
Incomplete resection	24 (38.1%)
Recurrence time, months	4.8 ± 3.4 (1–16)
Early recurrence	42 (66.7%)
ASM regimen adjustment	40 (63.5%)
Increased dosage	38 (60.3%)
Alternative therapy	11 (17.5%)
Combination therapy	17 (27.0%)

**Table 2 medicina-59-00785-t002:** Association between clinical variables and outcome in children with failed resection surgery.

Variable	Seizure Freedom (*n* = 23)		Seizure Remission (*n* = 26)		Good QoL (*n* = 39)	
	Statistics	*p* Value	Statistics	*p* Value	Statistics	*p* Value
Gender	0.01	0.94	0.01	0.94	0.81	0.37
Duration of epilepsy	−0.17	0.86	−0.34	0.74	−0.57	0.57
Seizure type	2.28	0.13	1.23	0.27	2.03	1.56
Surgery site	1.24	0.27	1.25	0.26	1.68	0.20
Incomplete resection	2.22	0.14	4.23	0.04 *	0.21	0.65
Early recurrence	5.79	0.02 *	5.53	0.02 *	7.57	0.01 *
ASM regimen adjustment	2.00	1.58	1.78	0.18	1.67	0.68
Increased dosage	1.00	0.32	0.77	0.38	0.08	0.78
Alternative therapy	0.13	0.72	1.08	0.30	1.53	0.22
Combination therapy	0.02	0.90	0.32	0.57	2.01	0.15

* indicates statistical significance between two compared groups (*p* < 0.05).

## Data Availability

Please see Appendix A.

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
