# Peer review of "Does Adjustment of Antiseizure Medication Regimen after Failed Epilepsy Surgery Improve Outcomes?"

_medicina, 2023, doi:10.3390/medicina59040785_

Round 1

Reviewer 1 Report

The authors explored which method of Antiepileptic drug regimen ad-52 justment was better at improving the seizure outcome and discussed subsequent possible treatment for children after the failure of initial surgery. This type of studies are necessary, because for clinicians, management of patients after the failure of initial surgery is a challenging medical problem. Although the study has some limitations considering the small number of patients, I think that, in general,  the text is clear and well-written.

I have few comments for the authors:

- General consideration: in the whole text (including the abstract) the words "antiepileptic drug (AED)" must be avoided: the correct words are: "antiseizure medication (ASM)"

- Methods: is the selection of patients in any way systematic or random? This should be detailed. Moreover, it is not mentioned what the reason was for the referral of the patients to the Department of Neurosurgery. Were all patients with pharmacoresistant epilepsy or is there another reason?

- Discussion: in the last part of this section, the authors should discuss more in depth the role of medical treatment in relationship with failed surgery: the authors should read and cite the paper by Ambrosi M et al. Expert Opin Pharmacother. 2017 Oct;18(14):1491-1498

Author Response

Reply to Reviewer 1

Dear Reviewer 1,

Thank you very much for your time involved in reviewing the manuscript and your very encouraging suggestions on the merits. Here below is our description on revision according to your comments.

Comment 1: The authors explored which method of Antiepileptic drug regimen adjustment was better at improving the seizure outcome and discussed subsequent possible treatment for children after the failure of initial surgery. This type of studies are necessary, because for clinicians, management of patients after the failure of initial surgery is a challenging medical problem. Although the study has some limitations considering the small number of patients, I think that, in general, the text is clear and well-written.

Response 1: Thanks to the reviewer for time taken in considering our manuscript and appreciation of the idea of the manuscript.

Comment 2: General consideration: in the whole text (including the abstract) the words "antiepileptic drug (AED)" must be avoided: the correct words are: "antiseizure medication (ASM)"

Response 2: We appreciate the reviewer’s recommendation and have corrected the phrasing as suggested. Thanks for the reviewer’s the time and effort, which we believe has resulted in a significantly improved manuscript.

Comment 3: Methods: is the selection of patients in any way systematic or random? This should be detailed. Moreover, it is not mentioned what the reason was for the referral of the patients to the Department of Neurosurgery. Were all patients with pharmacoresistant epilepsy or is there another reason?

Response 3: We thank the reviewer for bringing this important point to our attention. We recognize that we have not made our study patient selection very clear in our methods section. In brief, we included all children who had failed epilepsy surgery in our department between 2015 and 2021, all of whom had a preoperative diagnosis of refractory epilepsy. Now we have added more content about the patient selection of this study. We believe that such a change will make the structure of the paper clearer. Your comments and suggestions really helped us a lot. (Please see line 28, page 2 to line 6, page 3).

Comment 4: Discussion: in the last part of this section, the authors should discuss more in depth the role of medical treatment in relationship with failed surgery: the authors should read and cite the paper by Ambrosi M et al. Expert Opin Pharmacother. 2017 Oct;18(14):1491-1498

Response 4: Thank you for your valuable feedback on our manuscript. We appreciate your suggestion to discuss in more depth the role of medical treatment after surgery failure. We have read the paper by Ambrosi M et al and agree that it provides useful insights into out manuscript.

In light of this, we have revised the manuscript to include a more detailed discussion on the role of medical treatment. We have also cited the paper by Ambrosi M et al. to make the argument stronger. (Please see line 9, page 6 to line 3 page 8).

We tried our best to improve the manuscript and made some changes marked in revised paper which will not influence the content and framework of the paper. We appreciate for your warm work earnestly, and hope the correction will meet with approval. Once again, thank you very much for your comments and suggestions.

Sincerely,

Xuan Zhai

Reviewer 2 Report

The mansucript "Does Adjustment of Antiepileptic Drug Regimen after Failed 2 Epilepsy Surgery Improve Outcomes?" is interesting and well-written. The topic is important since children who undergo epileptic surgery have a relatively high risk of failure and recurrent seizures. However, the number of patients included in this study is small and the patients are heterogenous in many aspects and the most important being the treatment following failure. The results show that 23 patients (36.5%) continued theri previous treatment and 40 patients (65.5%) had AED regimen adjustment. When you like at Table 1 and the text below increased dosage, alternative therapy and combination therapy add up to much more than 40 patients. Why is that? Also later on in the manuscript you present only two groups of patients those with adjusted AED regimen and those without.

Moreover, what was the reason for AED regimen adjustment compared to no adjustment? How did you select the patients for AED regimen adjustment?

I was confused by the difference in % of patients with seizure-free rate following adjusted and non-adjusted AED regimen and the fact that it did not result in statistical difference. It is possible that the difference exists but was not detected because of the small sample size.

My opinion is that the discussion should be much more focused on AED treatment following failed epileptic surgery.

Author Response

Reply to Reviewer 2

Dear Reviewer 2,

On behalf of all the contributing authors, I would like to express our sincere appreciations of your constructive comments concerning our article. These comments are all valuable and helpful for improving our article. According to your comments, we have made extensive modifications to our manuscript. Point-by-point responses to you are listed below this letter.

Comment 1: The manuscript "Does Adjustment of Antiepileptic Drug Regimen after Failed 2 Epilepsy Surgery Improve Outcomes?" is interesting and well-written. The topic is important since children who undergo epileptic surgery have a relatively high risk of failure and recurrent seizures.

Response 1: We are grateful to the reviewer for their positive assessment and appreciation of our manuscript.

Comment 2: However, the number of patients included in this study is small and the patients are heterogenous in many aspects and the most important being the treatment following failure.

Response 2: We thank the reviewer for pointing out this important issue. The determination of the sample size for this study was based on the following considerations.

1). The case review in our hospital was based on the newly launched big data platform in the hospital, and the medical records after 2015 were more complete. Although the content of medical records prior to 2015 remains complete in the medical record system, the big data platform only includes a part of patients, and some patients cannot be retrieved. For the consideration of reducing selection bias, we always preferred to select patients admitted after 2015 for inclusion in the study.

2). Prior to the start of this study, we hypothesized an AUC value of 0.80 for the ROC curve of the regression model constructed by the included indicators. If the test level α=0.05 and the test efficacy (1-β)=0.8 were adopted, if the corresponding test efficacy was achieved. At this time, PASS11 can be used to estimate the minimum sample size of 38 cases. Due to sample size limitations, we only performed between-group comparisons and did not perform multivariate analyses. This was also done to ensure the reliability of the study results.

Ref#1: Hanley, J. A. and McNeil, B. J. 1983. 'A Method of Comparing the Areas under Receiver Operating Characteristic

PASS outcome:

3). As ASM treatment options after failed epilepsy surgery is an important but relatively under-discussed topic, there is not much previous literature on the subject and fewer references available. In contrast to other studies about failed epilepsy surgery, Lesley C Kaye et al. reviewed patients who underwent epilepsy surgery at the University of Washington Regional Epilepsy Center between 2007 and 2017 to explore factors influencing long-term seizure remission after failed surgery, the study had a sample size of 98 individuals. We included 283 children who underwent epilepsy resection surgery, with 63 children ultimately included in the analysis due to the relatively small number of seizures following epilepsy surgery.

Ref#2: Kaye LC, Poolos ZA, Miller JW, Poolos NP. Clinical factors associated with late seizure remission after failed epilepsy surgery. Epilepsy Behav. 2023 Jan;138:109055. doi: 10.1016/j.yebeh.2022.109055. Epub 2022 Dec 19. PMID: 36543042.

4). Of course, as we mentioned in the limitations. We were aware that our sample is small and that the study findings are not very convincing. However, for reasons of reducing selection bias, we did not continue to expand the time period included in the study. The novelty of this study is to investigate the influence of ASM adjustment after failed surgery, which can help to improve postoperative management. We consider therefore that our conclusion is statistically reliable enough to be published in this first report of the kind and expect that our subsequent prospective studies will prove and deepen the findings of this study.

For the second deficiency pointed out by the reviewer, we acknowledge the heterogeneity of this study, which is an inherent limitation of retrospective studies. The retrospective study approach was chosen to initially and superficially explore the impact of ASM adjustments in children with failed epilepsy surgery. Based on the results of this study, we propose to conduct a prospective cohort study to validate the results of this study and to further explore potential factors that may improve seizure outcomes and quality of life in children who have failed surgery. It is anticipated that the study could be completed within the next 3-5 years.

Comment 3: The results show that 23 patients (36.5%) continued theri previous treatment and 40 patients (65.5%) had AED regimen adjustment. When you like at Table 1 and the text below increased dosage, alternative therapy and combination therapy add up to much more than 40 patients. Why is that? Also later on in the manuscript you present only two groups of patients those with adjusted AED regimen and those without.

Response 2: We thank the reviewer for pointing out this lack of clarification, as this is a retrospective study, the median follow-up is 53 months. During the follow-up period, we would manage to improve seizure outcomes in children, and an individual with residual seizure after surgery may successively undergo dose increasing, alternative therapy and combination therapy, so it is not possible to simply divide children into 4 groups: continuing the current ASM therapy, increasing the dose, alternative therapy and combination therapy, as a patient will be included in more than one group, which is inherent to the limitations of retrospective real-world studies. For this reason, we chose a between-group comparison approach for our statistical analysis, comparing the outcomes of children with and without ASM adjustments, comparing the outcomes of children with and without dose increasing, and so on. We are aware that this was not well described in the manuscript and we have now added an explanation of this in the manuscript and hope that the protocol is now clearer. (please see 9 to 14, page 4)

Regarding the second issue indicated by the reviewer, we now discuss more in depth the role of medical treatment in relationship with failed surgery. (please see Response 6)

Comment 4: Moreover, what was the reason for AED regimen adjustment compared to no adjustment? How did you select the patients for AED regimen adjustment?

Response 4: We thank the reviewers for remaining cautious on this point, as we mentioned in the manuscript: “The ASM regimen is based on an individual medical basis (ASM already tried, seizure type, seizure frequency, age, adverse effects, ASM mechanism, patient preference and other factors) at the discretion of the neurologist.” Our medication management after a recurrence is individualized and the following are the key points we consider when adjusting ASM.

  1. All children who underwent epilepsy surgery at our centre were diagnosed with refractory epilepsy in the department of neurology preoperative, and they were on more than 2 types of ASM of the correct choice. We usually continued the pre-operative ASM regimen after surgery, so in most cases the ASM regimen is correct and appropriate in terms of the drug mechanism, indicated seizure types, and the synergistic effects.
  2. So if a child experienced surgery failure, our first consideration is whether the seizure type and interictal/ictal EEG pattern have changed after surgery. This is because, in clinical practice, many children with epilepsy have different sites of interictal epileptiform discharges(IEDs) and different seizure types after surgery, which may lead to ASM adjustments. If a child with epilepsy has a change in seizure type or interictal/ictal EEG pattern, the preoperative ASM may no longer be the best choice for him and we need to carry out a replacement of the type of ASM (alternative therapy in our manuscript). For example, if a child's seizures change from generalised to focal, sodium valproate may be replaced with Oxcarbazepine.
  3. If the child's seizure type and EEG presentation are the same as preoperatively, we will not consider alternative therapy, as this has been tried repeatedly by the neurologist before surgery.
  4. Then we will measure the ASM blood concentration. Because children are growing and developing, if their ASM regimen has not been adjusted, their ASM blood concentration may fall below the effective concentration due to weight gain. If the effective concentration is not reached, we will choose to increase the dose.
  5. If the child is already on more than 2 antiepileptic drugs and the dose is within the effective range, we will assess the child's current seizure frequency and the effect of seizure. We then weigh the drug load of dose increasing of ASM against the negative effects of seizure and choose to increase the medication or continue with the current ASM regimen, depending on the child's condition. We usually prefer to increase the dose if the current dose doesn’t reach the maximum dose, and if the maximum dose of a single drug is reached, we will opt for a combination of drugs.

These are the considerations we take into account when surgery failed, but the complexity of the situation in clinical practice makes it difficult to implement using simple criteria. For example, if at the outpatient clinic we decide that the seizure focus was not completely removed during surgery and there is a possibility of a second operation. Then we may not adjust the ASM, but actively prepare for a second surgical procedure, as we do not think this will necessarily provide a significant benefit while increasing the child's medication load.

We have therefore not described our criteria for ASM adjusting in the manuscript in such detail. Thank you again for your careful reading.

Comment 5: I was confused by the difference in % of patients with seizure-free rate following adjusted and non-adjusted AED regimen and the fact that it did not result in statistical difference. It is possible that the difference exists but was not detected because of the small sample size.

Response 5: We thank the reviewer for raising this important question, as noted by reviewer 2, the difference in the percentage of patients with seizure-free rates between the adjusted and non-adjusted AED regimens was not statistically significant in our study. This suggests that there may not be a significant difference between the two approaches in terms of seizure control. However, it is also possible that the difference exists but was not detected due to the small sample size. We prefer that there is no significant difference between the two groups because, with reference to previous studies, P < 0.05 means that the variable has a statistically significant impact on the outcome, whereas P > 0.05 does not completely exclude the effect of the variable on the outcome. And usually, 0.1 > P > 0.05 is considered as suspected impact factor. In this study, although the seizure-free rate was higher in children with adjusted ASM regimens than in those without, the P value was 0.16, and this value exceeds the category of suspected impact. So we think that adjustments to the ASM regimen are unlikely to improve postoperative outcomes in children with seizures. However, given the small sample size of this study, it might be prudent and necessary to discuss these data a bit more carefully. So we added more contents about this issue in the Discussion section and we appreciate the reviewer’s recommendation. (please see line 22, page 7 to line 3, page 8)

Comment 6: My opinion is that the discussion should be much more focused on AED treatment following failed epileptic surgery.

Response 6: Thank you for your feedback and suggestions regarding the discussion section of our paper.  We appreciate your valuable input. We agree that ASM treatment following failed epileptic surgery is an important aspect to consider and could have been discussed in more detail. In our paper, we mainly focused on the factors affecting the outcome after failed epileptic surgery. We acknowledge that a deeper discussion of pharmacological treatment after surgical failure would be more in keeping with the theme of this study and may add further value to the paper. We took your suggestion into consideration and updated our discussion section to include a more detailed analysis of the role of ASM in the management of children with epilepsy following failed surgical intervention. Thank you again for your feedback, which will undoubtedly help us to improve the quality of our paper. (Please see line 9, page 6 to line 3 page 8).

We would like to take this opportunity to thank you for all your time involved and this great opportunity for us to improve the manuscript. We hope you will find this revised version satisfactory.

Sincerely,

Xuan Zhai

Round 2

Reviewer 2 Report

No further comments